



# North Atlantic Oscillation (NAO) in the Paleoclimate Modelling Intercomparison Project (PMIP)

Anni Zhao[1], Chris Brierley[2], Venni Arra[2], Xiaoxu Shi[3], and Yongyun Hu[1]

[1]Department of Atmospheric and Oceanic Sciences, School of Physics, Peking University, Beijing, 100871, China
[2]Department of Geography, University College London, London, WC1E 6BT, UK
[3]Southern Marine Science and Engineering Guangdong Laboratory, Zhuhai, 519082, China

**Correspondence:** Yongyun Hu (yyhu@pku.edu.cn)

**Abstract.** The North Atlantic Oscillation (NAO) is one of the main modes of climate variability and the dominant mode of large-scale atmospheric variability in the North Atlantic basin and has large impacts on the European climate, whose future behaviour remains uncertain. Here we assess the NAO response in past and future climates by looking at a comprehensive set of coupled model simulations performed by the Paleoclimate Model Intercomparison Project (PMIP) and the Coupled Model Intercomparison Project (CMIP) for four experiments: the mid-Holocene (6 ka; *midHolocene*), the Last Glacial Maximum (21 ka; *lgm*), the last interglacial (127 ka; *lig127k*) and an idealised future warming scenario with abrupt quadrupled $CO_2$. Although there are various setups across experiments, the *midHolocene* and *lig127k* are mainly characterised by altered orbital configurations, inducing variations in the seasonal cycle, and the *lgm* and *abrupt4xCO2* are mainly characterised by various GHG forcing that induces great global temperature change. Our results show that the NAO is sensitive to GHG-forcing-induced temperature changes but not the orbital configurations. NAO weakens in response to cooling and strengthens to warming. The associated teleconnections change consistently with the theory and are sensitive to the change in NAO amplitude. The two orbital experiments do not show a clear change in associated temperature and precipitation. The weakened NAO in the *lgm* is associated with a cooler and drier northern Europe, while the enhanced NAO in the *abrupt4xCO2* causes a warmer and wetter northern Europe as compared to the *piControl*. No clear relationship is found in the ENSO-NAO teleconnection.

## 1 Introduction

The North Atlantic Oscillation (NAO), characterised by seesaw sea level pressure anomalies between the Icelandic Low and the Azores High, represents one of the main modes of climate variability and the dominant mode of large-scale atmospheric variability in the North Atlantic basin (Hurrell et al., 2003). It affects climate throughout all seasons, most pronounced during boreal winter (Walker and Bliss, 1932). There is ongoing debate on the relationship between the NAO and the North Annular Mode (NAM, also known as the Arctic Oscillation): some studies interpret the NAM as a statistical artefact of two independent regional modes (NAO and the Pacific-North American pattern), and thus viewing the NAO as a stand-alone mode (Ambaum et al., 2001); while others view the NAO as a regional expression of the NAM (e.g. Thompson and Wallace, 1998; Kimoto et al., 2001) and do not distinguish between the two. The latest phase of the Intergovernmental Panel on Climate Change (IPCC AR6)





often considers the NAM and the NAO as the same entity, as the NAO can explain most of the variance in zonally-averaged

hemispheric circulation and exhibits a high correlation in associated time-series (IPCC, 2021b).

NAO is predominantly driven by the internal atmospheric variability of mid-latitude dynamics (Lorenz and Hartmann, 2003) related to the interaction between eddy-mean flow (Feldstein and Franzke, 2017; Kimoto et al., 2001) and the coupling between troposphere and stratosphere (Wang and Ting, 2022; Omrani et al., 2022), and the coupling among atmosphere, land, and ocean (Marshall et al., 2001). Various mechanisms play important roles in shorter to longer-term NAO variability (Stephenson et al.,

2006), including sea surface temperature (SST) (Mosedale et al., 2006; Hurrell et al., 2004), greenhouse gas (GHG) forcing (Stephenson et al., 2006; Barnes and Polvani, 2013), sea ice (Pedersen et al., 2016; Smith et al., 2017), and snow cover changes (Henderson et al., 2018; Spencer and Essery, 2016). The NAO strongly impacts regional surface climate variables (Woollings et al., 2015), notably modulating surface air temperature over Europe (Hurrell, 1995). The interplay between the two pressure systems creates a pressure gradient that influences the strength and direction of westerly winds and storm tracks across the

North Atlantic. The phases of NAO significantly influence the temperature and precipitation variability over the North Atlantic and Eurasia, as well as related westerly winds and storm tracks (Woollings et al., 2014). In its positive phase, the NAO is formed by relatively stronger low pressure over Iceland in the mid to high latitudes and stronger high pressure over the Azores in the subtropics, leading to an enhanced and northward shift of the jet stream, strengthened trade winds, and a northward shift of extra-tropical storms. The positive NAO typically results in a warmer and wetter northern Europe and a colder and drier

Mediterranean. Conversely, the negative phase of the NAO shows the reverse change in pressure systems, leading to a wavier jet stream that extends southward and a southward shift of storms, resulting in a colder northern Europe and a warmer southern Europe.

Research around changes in the NAO involves some awkward terminology. Here we consider the NAO to be a mode of variability, which can be described by the interannual changes in the NAO index. Changes in the mean state of the Icelandic

Low and Azores High will also project onto the pattern of that mode of variability - leading to secular shifts in the NAO index (i.e. shift in the mean state of the NAO), even if the oscillation itself remains the same. These are easy to distinguish in equilibrated climate model experiments, such as those used here. However, it is harder when looking at continuous NAO indices that capture a large range of frequencies, such as from palaeoclimate reconstructions or long transient simulations.

Several studies have carried out proxy-based reconstructed NAO (e.g. Vinther et al., 2003; Cook et al., 2019), though the

reconstructions are limited by significant uncertainties arising from the limitations of archives and chronologies (Hernández et al., 2020), diverse methodologies (Michel et al., 2020), or the time scale effect on NAO variability (Woollings et al., 2015). For example, a positive NAO-like configuration was suggested during the mid-Holocene (6 ka) based on reconstructed terrestrial temperature over Europe (Mauri et al., 2014) and SSTs (Rimbu et al., 2003). Earlier studies have explored the NAO in paleoclimate simulations; however, the exact behaviour of NAO remains uncertain. The mid-Holocene simulations of the

Palaeoclimate Modeling Intercomparison Project 2 (PMIP2) show various changes in NAO variability as compared to the preindustrial condition, as reduced (Lü et al., 2010), no change (Otto-Bliesner et al., 2006), or enhanced (Gladstone et al., 2005), the results of the PMIP3 are also model-dependent (Găinuşă-Bogdan et al., 2020). Under the cooler-than-present condition of the Last Glacial Maximum (LGM; 21 ka), simulations show that the NAO pattern and variability weaken drastically in





simulations, and the centres of the NAO pattern shift southward compared to both the pre-industrial and the mid-Holocene (Lü
et al., 2010; Otto-Bliesner et al., 2006).

Future simulations predict a more positive mean state of the NAO index, although the response is highly model dependent
(Miller et al., 2006; Osborn, 2004; Bader et al., 2011). The mean state of the NAO index is projected to shift positively under
high-emission scenarios while showing less robust change under low-emission scenarios by the end of the 21st century, with
a large diversity across the Coupled Model Intercomparison Project (CMIP) simulations (Lee et al., 2021). The large spread
is partly due to the contrasting influences of the negative phase of the NAO induced by Arctic amplification (Harvey et al.,
2015; Peings et al., 2017; Screen et al., 2018) and the positive phase associated with enhanced warming in the tropical upper-
troposphere (Vallis et al., 2015) in particular models in response to anthropogenic forcing (Harvey et al., 2014, 2015; Vallis
et al., 2015; Oudar et al., 2017).

In this study, we look at simulated NAO change by the new generation of climate models for three PMIP experiments and
a CMIP $CO_2$ scenario experiment to assess NAO change through time. The PMIP experiments represent a colder LGM and
two past climates with altered orbital configurations, the last interglacial (LIG; 127 ka) and the mid-Holocene. The CMIP $CO_2$
scenario experiment is an idealised warming projection designed with abrupt quadrupled $CO_2$. Sect. 2 introduces the models,
experimental design, and analysis methods. A comparison between the *piControl* experiment with observation is provided
in Sect. 3 for model evaluation. Sect. 4 describes changes in the mean state relative to the *piControl*. Sect. 5 presents the
description of the change in NAO and the associated discussion. NAO teleconnections change, and discussion is given in Sect.
6. A brief conclusion is provided at the end as Sect. 7. Comparison between PMIP simulations with proxy data is not the aim of
this study, and it has been done in assessing single experiments (see Brierley et al., 2020; Otto-Bliesner et al., 2021; Kageyama
et al., 2021), so it will not be discussed in detail below.

## 2  Methods

This study involves a bunch of simulations across 2 CMIP/PMIP generations, 5 experiments and 34 climate models (Table 1).
Studies on individual experiments (Brierley et al., 2020; Otto-Bliesner et al., 2021; Kageyama et al., 2021) and single-model
evaluations have been done by model groups (e.g. Otto-Bliesner et al., 2020). Therefore, we will keep the description of models
and experimental design brief here, whilst references will be provided to find details.

### 2.1  Models

State-of-the-art coupled global climate models represent the climate system through physical equations based on the physical,
chemical and biological properties of the system's components, their interactions and feedback, and other known properties.
Climate models are tools to investigate how the climate responds to various climate forcings and project future climate change
in different scenarios. The CMIP has become a significant international multi-model research activity to investigate the Earth
system response to forcing, evaluating the state-of-the-art coupled global climate models and assessing future climate changes
and their uncertainties in scenarios (Eyring et al., 2016; Taylor et al., 2012). Simulations from the PMIP look at the past and



serve as out-of-sample tests to study the roles of forcings and their feedbacks that establish the palaeoclimate and to evaluate the performance of models that are used to project future climate changes (Kageyama et al., 2018). Here, we choose the models that have completed the *piControl* simulations and at least one of the PMIP simulations (see Sect. 2.2) and have provided monthly surface temperature, precipitation and sea level pressure (SLP) for at least 30 years for all the simulations. Table 1 lists the 34 chosen models and their participation in the six experiments (see Sect. 2.2). Further details of the model design can be found in the listed references in Table 1 here and Table 9.A.1 of Flato et al. (2013) for the CMIP5 models and Table AII.5 of IPCC (2021a) for the CMIP6 models.

## 2.2 Experimental design

This study uses simulations from six experiments with the protocols defined under the CMIP (Taylor et al., 2012; Eyring et al., 2016) or the PMIP (Kageyama et al., 2018): a pre-industrial control, one idealised $CO_2$-forced future warming scenario and three past climate experiments which include a colder *lgm* and two altered orbital configurations *midHolocene* and *lig127k*. Model participation in each experiment is detailed in Table 1.

The DECK *piControl* was performed by all selected models, employing coupled atmosphere-ocean and boundary conditions approximately constant to 1850CE (Eyring et al., 2016). Depending on the individual model design, aerosol, vegetation, and ice sheet configurations are either interactive or prescribed as modern conditions. The CMIP6 *piControl* uses more realistic GHG concentrations and a lower solar constant compared to its earlier generation, the CMIP5. The *piControl* serves as a baseline for assessing changes in other experiments. Model groups run their *piControl* simulations for a few hundred to a few thousand model years to reach an equilibrium state.

The idealised $CO_2$-forced experiment, *abrupt4xCO2*, is also a requirement of the DECK CMIP experiments (Eyring et al., 2016). It is designed to reveal the basic feedback response of a model to high GHG forcing (Eyring et al., 2016), by having the atmospheric $CO_2$ concentration suddenly and immediately quadruple relative to the 1850 CE condition prescribed in the *piControl* at the beginning and then maintaining constant (Eyring et al., 2016). The *abrupt4xCO2* simulations enable the estimation of a model's equilibrium climate sensitivity (Zelinka et al., 2020). Its protocol remains unchanged as in the CMIP5, facilitating comparisons across CMIP generations.

The *lgm* experiment is designed to examine the effect of altered ice sheets, continental extent due to reduction in sea level, and lower GHG forcings (Kageyama et al., 2018). The major difference between the PMIP3 and PMIP4 *lgm* simulations is the choice of the prescribed ice sheets. While the PMIP3 simulations all used the same composite ice-sheet reconstructions(Abe-Ouchi et al., 2015), the PMIP4 simulations chose one of the three as summarised in Kageyama et al. (2021): the original PMIP3-CMIP5 ice sheet (Abe-Ouchi et al., 2015), GLAC-1D (Ivanovic et al., 2016) and ICE-6G_C (Peltier et al., 2015; Argus et al., 2014). Although these reconstructions have similar ice sheet extent, they differ in the heights of the Laurentide, Fennoscandian, and West Antarctica ice sheets.

The *midHolocene* and *lig127k* experiments are designed to examine the climate system responses to orbital forcings that differ from the pre-industrial (Kageyama et al., 2018). The experimental designs for the two experiments are outlined by Otto-Bliesner et al. (2017). The *midHolocene*, included since the beginning of the PMIP (Joussaume and Taylor, 1995; Braconnot



et al., 2000), set to an orbital configuration at 6 ka following Berger (1978) and Berger and Loutre (1991). The orbit at 6
     ka was characterised by larger obliquity, and its perihelion occurred near the boreal autumn equinox. The prescribed GHG
     concentrations in the PMIP4 *midHolocene* use more realistic concentrations derived from ice cores and observations (Otto-
     Bliesner et al., 2017) than the PMIP3, which contributes to a decrease of 0.3 W m$^{-2}$ in effective radiative forcing (Otto-
     Bliesner et al., 2017). The *lig127k* uses the orbital configuration at 127 ka (Berger and Loutre, 1991), characterised by a larger
eccentricity than the pre-industrial and its perihelion close to the boreal summer solstice. Both periods were characterised
     by receiving more insolation at the top of the atmosphere (TOA) in the Northern Hemisphere during JJA and less incoming
     solar radiation in both hemispheres during DJF compared to 1850 CE, with the LIG having stronger orbital forcing than the
     mid-Holocene.

## 2.3    Analysis and definitions

The analysis in this work follows the workflow described by Zhao et al. (2022). We create a curated replica of the relevant
     simulation outputs available on the System Grid Federation (ESGF; Balaji et al., 2018), which stores the CMIP6 outputs in a
     standardised format (Juckes et al., 2020). Digital Object Identifier (doi) for each simulation downloaded from the ESGF can be
     found in the Supplement. We also require some simulations from the model groups if they are not available on the ESGF. Then
     we apply calendar adjustment on the *midHolocene*, *lig127k* and *lgm* monthly output that re-aggregate the monthly output from
a present-day calendar to those representing the past via the PaleoCalAdjust software by Bartlein and Shafer (2019). In order
     to compare with equilibrium simulations, we take the last 50 model years of the *abrupt4xCO2* simulations in analysis. The
     Climate Variability Diagnostics Package (CVDP; Brierley and Wainer, 2018) had been modified for palaeoclimate purposes
     (Brierley and Wainer, 2018). We run the modified CVDP on each simulation to calculate multiple pertinent time series and
     spatial fields. The CVDP outputs have been regridded to 1° by 1° before analysing to eliminate the effects of different initial
resolutions across all the models. Multi-model mean anomalies take the average of regridded anomaly (*experiment - piControl*)
     across the ensemble members (Brierley and Wainer, 2018). Please see Zhao et al. (2022) for more details. We evaluate ensemble
     consistency as at least two-thirds of the models agree on the sign of the multi-model mean. Results of the PMIP4 and PMIP3
     have not shown fundamental differences in statistics (Brierley et al., 2020; Kageyama et al., 2021), so we, therefore, combine
     the two indistinguishable phases in the purpose of enlarging the size of ensembles, which have been adopted in analysing
ENSO (Brown et al., 2020) and Indian dipole (Brierley et al., 2023) across experiments.

     NAO can be defined based on the principal component (PC-based) as the leading empirical orthogonal function (EOF) of
     SLP variance during boreal winter over the North Atlantic-Europe sector (IPCC, 2021b), or based on observations from fixed
     locations as the difference of normalised SLP anomalies between stations located in the Azores-Ponta Delgada and Iceland
     (Jones et al., 1997) or the difference between London and Paris (Cornes et al., 2013). Here, we adopt the definition of NAO as
the leading empirical orthogonal function (EOF) of DJF SLP anomalies over the Northern Hemisphere north of 20°N to 80°N
     and 90°W to 40°E (hereafter referred to as the NAO pattern). The NAO index is the normalised PC time series of the EOF.
     Instead of using the NAO index, we use the amplitude of NAO to show its strength, which is measured as the standard deviation
     of the renormalised PC time series (i.e. the NAO index times the standard deviation of the spatial NAO pattern over the region).





We also present spatial patterns associated with the NAO by linearly regressing the monthly temperature and precipitation onto

the NAO index to evaluate the association between the NAO and surface climate variables (Brierley and Wainer, 2018). We also examine the similarity between the NAO and the NAM in the PMIP experiments. All the experiments show that NAM and NAO are strongly correlated in the time series of the NAO indices, explained by variance and amplitude (not shown). Therefore, the following sections do not distinguish the NAO from the NAM.

## 3   Comparison with observations

Before looking at the response in experiments, we first evaluate the ability of the CMIP models used in this study to reproduce climate states during the pre-industrial period by comparing the *piControl* simulations with the C20 Reanalysis dataset (Compo et al., 2011), as shown in Figure 1. Models generally capture the spatial pattern of observed mean DJF temperature, but bias in producing too cold Arctic, especially over Greenland and Nunavut (Figure 1a, b). The cold bias has been detected by earlier PMIP analyses on a single experiment (Brierley et al., 2020; Otto-Bliesner et al., 2021), both showing a large spread

across the models in Arctic regions between 60°N and 90°N. The bias is partly associated with the model's representation of cloud radiative processes (Zhang et al., 2023) and atmospheric dynamics (Hall et al., 2021) and may also be affected by Arctic hydrography (Khosravi et al., 2022). Figure 1c and d show that models capture the DJF mean precipitation pattern over the North Atlantic Ocean, though they underestimate the amount of precipitation and shift the maximum location westward slightly as compared to the observation. The simulated SLP generally agrees with the observation (Figure 1 e and f), although

it produces lower pressure along the coast of Greenland.

    The NAO is the leading EOF of DJF SLP over the North Atlantic-Europe sector (see Sect. 2.3), whose simulated pattern shows a positive phase during the pre-industrial (Figure 1g, h), consistent with observations but with significant negative anomalies extended further into the centre of Greenland and an underestimation of positive anomalies around 30-40°N North Atlantic Ocean. In the *piControl* simulations, the NAO explains 41.5% (34.3%-52.0%) of the total variance, slightly less than

42.3% from observation. The *piControl* surface air temperature imprint of the NAO shows a pattern of warmer mid-latitudes and colder subpolar and subtropical regions than the observation (Figure 1i and j), with the effect of a stronger jet stream located further north, stronger trade winds and northward shift of extratropical storms associated with a positive phase of NAO IPCC (2021b). The related northward shift of storm tracks with positive NAO phases results in a wetter northern Europe, a drier southern Europe and a northward shift of precipitation over the ocean. Both observations and models capture the pattern

(Figure 1k, l), but the simulated pattern is smoother than the observation with underestimated change magnitudes.

## 4   Change in mean state

First the changes in mean climate over the North Atlantic are evaluated for each experiment, relative to the *piControl* simulations. Figure 2 summarises the DJF response of the experiments to the two forcings for SAT (left column), precipitation (middle column), and SLP (right column). We present and discuss the response of *midHolocene* and *lig127k* experiments to-



gether firstly to show the climate response to orbital forcing, and then present those of the *lgm* and *abrupt4xCO2* experiments for response to GHG forcing, though the *lgm* experiment also involves the response to the ice sheet.

The orbit configuration was characterised by larger obliquity at 6 ka and larger eccentricity at 127 ka than pre-industrial ones. The perihelion at 6 ka occurred near the boreal autumn equinox. At 127 ka, it occurred close to the boreal summer solstice rather than near the boreal winter solstice during the pre-industrial (Berger and Loutre, 1991). Both periods were

characterised by receiving less incoming solar radiation globally during DJF compared to 1850 CE, with a stronger reduction at 127 ka, suggesting an expectation of cooling in both experiments. The meridional SAT gradients of the *midHolocene* and *lig127k* experiments are weaker than the *piControl*, and the *lig127k* displays a stronger signal (Figure 2a, d). The DJF mean temperature change averaged over the N. Atlantic sector in the *midHolocene* experiment is 0.07°C cooler than the *piControl*. The cooling is stronger in the *lig127k* experiment at -0.52°C. Both experiments produce polar amplification, though the signal

is inconsistent across the ensembles. The *lig127k* simulations show a large spread across the models in producing Arctic warming, and the *midHolocene* simulations show a larger ensemble spread over the whole region of the North Atlantic sector. The Arctic warming during the boreal winter suggests other important processes, e.g. polar amplification with smaller and thinner sea ice in the Arctic (Serreze and Barry, 2011) and ocean memory (Marino et al., 2015; Govin et al., 2012).

Meanwhile, both experiments do not incorporate some regional processes that can cause regional coolings, e.g. the *lig127k*

simulations did not include meltwater from ice sheets over Scandinavia and Canada related to the cooling in the Nordic Seas and south of Greenland shown by LIG reconstruction (Barlow et al., 2018; Otto-Bliesner et al., 2021). Earlier studies have compared the experiments with reconstruction and have confirmed that both experiments underestimate the Arctic warming (Brierley et al., 2020; Otto-Bliesner et al., 2021). The underestimated mid-Holocene Arctic warming has existed since the PMIP3-CMIP5 (Harrison et al., 2015; Yoshimori and Suzuki, 2019). The regional temperature bias in the CMIP5 *midHolocene*,

*piControl* and *historical* are similar (Ackerley et al., 2017; Harrison et al., 2015), which indicates the persistent errors in representing the climate system. The Euro-Atlantic precipitation fields show a drier Europe and an apparent drying off of the North American east coast in both experiments as compared to the *piControl* (Figure 2b, e). There is a wetting over the rest of the Atlantic in the ensemble average, although the models do not all agree on the sign of the change. The signal of multi-model mean precipitation change is affected by NorESM2-LM, which produces a much larger precipitation increase north of 40°N

than other models in both experiments. Figure 2c, f shows that SLP increases over lower latitudes and decreases over high latitudes.

The LGM was characterised by a colder climate with a global mean cooling of 5–7°C (Gulev et al., 2021) and large ice sheets covering the mid- to high-latitudes in the Northern Hemisphere (Clark and Mix, 2002; Clark et al., 2009). The averaged DJF cooling over the North Atlantic sector is -9.86°C in the *lgm* simulations compared to the *piControl*. The *lgm* simulations

show consistent cooling over the sector with a greater temperature decrease over land than over sea (Figure 2g). The greatest cooling is found over the Norwegian and Barents Seas and parts of Scandinavia and North America, which reflects the change in surface height and albedo related to ice sheet change (Kageyama et al., 2021). The cooling is also partly affected by the advection of the cold temperature anomalies downwind of the ice sheets (Kageyama et al., 2021) and is partly related to a weakened Atlantic meridional overturning circulation (AMOC) (Jonkers et al., 2023). Though the cooling signal is consistent,





the ensemble shows a large spread in simulating the magnitude of cooling affected by the choice of ice sheets of the PMIP3
simulations shown in Kageyama et al. (2021). Shi et al. (2023) decomposed the contribution of individual boundary conditions
and forcings to the LGM cooling and found that the cooling over the sea within the region is mainly due to the GHG forcings
and that over the continent is due to changes in ice sheets. The *abrupt4xCO2* simulations show a contrasting signal to the
LGM by showing consistent warming over the whole North Atlantic region with greater warming over land (Figure 2j). The

*abrupt4xCO2* experiment shows a warming of 6.41°C over the sector. The weakest warming appears over the ocean south
of Iceland, where the AMOC condenses. Earlier studies suggest the AMOC weakening in the *abrupt4xCO2* experiment in
response to warming (Zhu et al., 2023; Madan et al., 2024). The *lgm* simulations show enhanced precipitation over mid-
latitudes North Atlantic Ocean as compared to the *piControl*, but the signal is not consistent across the ensemble (Figure 2h).
The switch of precipitation change signal represents a northward shift of the jet stream (Figure 2i) relative to the *piControl*,

whereas the inconsistency shows the uncertainty in producing the location of the jet stream. Though the increase in the multi-
model mean of *abrupt4xCO2* precipitation change is consistent, the spatial pattern of the change by individual models is
complex. The weakened precipitation south of Iceland is inconsistent across the *abrupt4xCO2* ensemble (Figure 2i), as about
half of the models produce such a reduction. The SLP increases uniformly in the *lgm* simulations, consistent with the enhanced
SAT cooling (Figure 2g). The higher SLP over the polar regions, coupled with a cooler and dryer climate, is related to the large

Laurentide ice sheet (Otto-Bliesner et al., 2006). The pattern of *abrupt4xCO2* SLP change (Figure 2l) is not consistent with
temperature change. The inactive prescribed Greenland ice sheets in simulations (Eyring et al., 2016) cause the SLP increase
over the region.

## 5   Change in NAO

In the *piControl* simulations, the NAO explains 41.5% of DJF SLP variance and shows a distinguishable positive phase with

a strong negative SLP centre over the northeast of Iceland and a positive centre over the northeast of the Azores (Sect. 3).
Notably, 9 out of the 34 models display an eastward shifted positive centre over Europe along with varying degrees of pattern
strength, and UofT-CCSM-4 produces dipole positive centres over the Mediterranean and the central USA. The DJF NAO time
series has an average amplitude of 2.27 hPa in the *piControl* simulations, ranging from 0.93 hPa (CSIRO-Mk3-6-0) to 2.99 hPa
(CCSM4).

The NAO patterns of the two orbital experiments are not distinguishable from the *piControl* (Figure 3a, c). The similarity
in the NAO pattern between the *midHolocene* and the *piControl* agrees with the findings of earlier modelling studies (Otto-
Bliesner et al., 2006; Gladstone et al., 2005; Stephenson et al., 2006). It is also consistent with the PMIP2 simulations (Lü
et al., 2010) that found the mid-Holocene leading EOF of DJF SLP is similar to the PI. Models produce complex changes
in the NAO pattern in both experiments and show a spread south of Greenland and Iceland near 50°N (Figure 3b, d). Of the

32 models, 11 display a clear weakening and 7 show an enhanced positive NAO phase in the *midHolocene*, respectively, as
compared to the *piControl*; a similar result of spatial change occurs for the *lig127k* experiment, as 6 (5) out of the 14 models
display weakness (enhancement) in the *lig127k* as compared to the *piControl* (not shown). The average percentage of explained





variance during the *midHolocene* is 42.4% (33.2-51.8%), whose range is close to the *piControl* (Figure 4). No clear changes in explained variance stay consistent with Gladstone et al. (2005), which, in conjunction with inter-model inconsistencies, leads

to uncertainties about the mid-Holocene NAO behaviour. Changes in explained variance vary across the ensemble (Figure 4b), ranging from -7.1% by EC-Earth-2-2 to 10.1% by UofT-CCSM-4. In the *lig127k* simulations, models display an averaged explained variance at 43.4% (38.3% to 47.7%), different from the 42.7% (35.7-52.0%) in the corresponding *piControl* simulations. Changes in explained variance by individual models vary across the ensemble and do not show a trend (Figure 4b). The NAO amplitude in the *midHolocene* simulations is 2.31 hPa (0.77-3.15 hPa). The range and distribution of the *midHolocene*

NAO amplitude are similar to those of the *piControl* (Figure 5), though the differences between *midHolocene* and *piControl* by individual models range from -0.30 hPa (EC-Earth3-LR) to 0.79 hPa (CSIRO-MK3-6-0). Though 19 out of 32 models present increased NAO amplitude, the ensemble mean change in the NAO amplitude is 0.06 hPa. The slight increase is largely affected by CSIRO-MK3-6-0, UofT-CCSM4 and HadGEM2-CC, which contribute an intensification at 0.79 hPa, 0.50 hPa and 0.43 hPa, respectively. The *lig127k* NAO amplitude is 2.18 hPa, similar to the *piControl* at 2.28 hPa with a more concentrated

distribution (Figure 5). The results of the two orbital-driven experiments suggest that the NAO response is unaffected by the seasonal variation induced by altered orbital configurations.

In the *lgm* experiment, the NAO weakens than the *piControl* (Figure 3e). There is a southward shift of both the negative and positive centres of the NAO as compared to the *piControl* (Figure 3e), though with a large model spread over the N. Atlantic Ocean between 40°N and 65°N (Figure 3f). The weakened NAO and centre shifts in the *lgm* are consistent with the

results of Lü et al. (2010) and Otto-Bliesner et al. (2006). Models show large variations in producing the location of positive and negative centres of the NAO. 8 out of the 13 models display the switch of signal between 50°N and 60°N, where IPSL-CM5A-LR displays its negative centre while COSMOS-ASO shows its positive centre. The NAO explains 39.6% (33-48.5%) of SLP variability in the *lgm* (Figure4a) that is -1.6% less than the corresponding *piControl* simulations. The majority of the models show a reduction in explained variance (Figure 4b), except FGOALS-g2 (9.2%), MIROC-ES2L (8.7%) and INM-

CM4-8 (3.3%). The *lgm* NAO amplitude is 1.58 (1.02-2.55) hPa, which is 0.59 hPa weaker than the *piControl* (Figure 5). UofT-CCSM-4 contributes to the largest weakness in amplitude at -1.61 hPa. Only INM-CM4-8, MIROC-ES2L and MIROC-ESM contribute to enhanced amplitude in the *lgm* simulations. Quartiles and median in Figure 5 show a general reduction in *lgm* amplitude than the *piControl*, which implies a weaker NAO in the *lgm*. The weakness is partly related to the presence of the Laurentide and Scandinavian ice sheets (Braconnot et al., 2007). The *abrupt4xCO2* experiment shows the opposite change

in the NAO by showing enhancement and a slight northward shift in centres than the *piControl* (Figs 3g,h). Models produce increased explained variance (Figure 4) and stronger amplitude (Figure5) than the *piControl*. The *abrupt4xCO2* simulations explain 43.2% (32.9-52.6%) of the SLP anomalies, which are 2.4% (-12.9% to 12.5%) larger than the corresponding *piControl* simulations. Only 6 out of 25 models produce a reduction in explained variance, in which MIROC-ESM presents the largest reduction by -12.9%, even larger than that in the *lgm* simulation at -4.7% as compared to the *piControl*. The NAO amplitude

in the *abrupt4xCO2* experiment is 2.28 hPa on average (ranging from 1.12 to 3.00 Pa). Though it is 0.11 hPa higher than the corresponding *piControl*, the median of *abrupt4xCO2* NAO amplitude is slightly lower than that of the corresponding *piControl*





(Figure 5. Overall, the NAO is sensitive to GHG forcing and temperature change, as the cooling weakens the NAO and the warming strengthens it, which implies that GHG changes in the future might play a role in modulating NAO behaviour.

The NAO is a variation in the atmospheric pressure differences between the Icelandic Low and the Azores High. As described
in Sect. 1, the NAO variability is primarily driven by the internal variability of mid-latitude atmospheric dynamics (Lorenz and Hartmann, 2003). The findings of Rind et al. (2005a, b) show that tropospheric and stratospheric climate changes influence the NAO response via altering propagating waves and angular momentum transport and changes in planetary wave energy. SST (Mosedale et al., 2006; Hurrell et al., 2004), GHG forcing (Stephenson et al., 2006; Barnes and Polvani, 2013), sea ice (Pedersen et al., 2016; Smith et al., 2017) and snow cover changes (Henderson et al., 2018; Spencer and Essery, 2016) could
also be potential drivers. Rind et al. (2005b) stated that the AO response during the Ice Age was dominated by changes in the eddy transport of sensible heat and local high-latitude forcing. Lü et al. (2010) examined the drivers of the NAO variability in the PMIP2 *midHolocene* and *lgm* experiments and found that the upward-propagating stationary Rossby waves might lead to NAO amplitude weakening. Based on four simulations, their *midHolocene* experiment presented a slight reduction in the AO intensity. Our results show the opposite change as the NAO amplitude is slightly strengthened, but it is affected by a few
models that produce a large enhancement. Our results of the *lgm* and *abrupt4xCO2* experiments agree with the findings of Rind et al. (2005a, b) as the NAO changes positively in global warming experiments and changes negatively in cooling experiments. They stated that the NAO index change is related to the eddy angular momentum transport change. The weakened *lgm* NAO amplitude is possibly due to cooler SST and reduced atmospheric moisture content as well as a shift of stationary waves and a weakened polar vortex linked to wave-mean flow interaction, and thus a reduction in polar westerlies (Lü et al., 2010).

## 6    Remote effects

A positive NAO in the *piControl* is associated with warmer and wetter northern Europe and central North America and colder and drier Mediterranean and northern North America (Figure 1j, l). Compared with the *piControl*, the *midHolocene* experiment shows a warmer continent associated with the NAO (Figure 6a), with the warming extending farther into central Eurasia and central North America (not shown). Differently, the *lig127k* shows cooling over northern North America and central Europe
than the *piControl* (Figure 6c). The DJF mean precipitation associated with the NAO increases over the North Atlantic Ocean north of 30°N and over northern Europe, while decreases over central Europe and North America in the *midHolocene* (Figure 7a). The *lig127k* experiment shows a similar pattern but with less precipitation over east of 20°W and north of 50°N relative to the *piControl* (Figure 7c) However, mean changes in the DJF mean temperature and precipitation associated with the NAO are not robust in the *midHolocene* and *lig127k* as compared to the *piControl*, and the changes are not consistent across the
ensembles (Figure 6a-d and 7a-d). The mid-Holocene models capture changes in the mean state (Fig. 2a) that project onto the positive phase of the NAO (Fig. 1h), as reconstructed by Funder et al. (2011). Proxy reconstructions also suggest that regional temperature patterns over Europe were primarily forced by a positive NAO-like phase during the mid-Holocene as opposed to radiative responses forced by changes in the seasonal insolation cycle (Rimbu et al., 2003; Mauri et al., 2014). However, none





of these papers include archives with sufficiently high temporal resolution to compare with the inter-annual variability being

investigated here.

The temperature and precipitation patterns associated with the NAO shift southward in the *lgm* experiment (Figures 6e and 7e), which suggests a weakness of features of the positive boreal winter NAO. Associated with the NAO, the *lgm* experiment presents colder and drier central North America and northern Europe and warmer and wetter Mediterranean and northern North America in the *lgm* than the *piControl*, with the patterns extending to central continents (not shown). As the strengthened precipitation teleconnection pattern over the western Iberian Peninsula is also present in the mean-state change (Figure 2h), it suggests that the strengthened Mediterranean precipitation pattern is not necessarily directly forced by the NAO. In the *abrupt4xCO2* experiment, opposite to the *lgm*, features of the positive boreal winter NAO are enhanced than the *piControl* (Figures 6g and 7g). The associated changes in temperature and precipitation do not match the pattern and magnitude of robust change in mean state temperature and precipitation (Figure 2), which implies that changes less influence change in mean state climatology in the NAO and have other attributions. For example, the large ice sheets during the LGM limited turbulent air-sea heat fluxes in the high-latitude North Atlantic that influenced the NAO and cooled the temperature (Sect. 4). The warming in the *abrupt4xCO2* can be explained by the thermodynamic processes affecting heat and moisture transport without changing the large-scale atmospheric circulation patterns (Stephenson et al., 2006). The enhanced warming of the surface boundary layer enhances vertical latent heating, which in turn influences the atmospheric moisture content and baroclinic stability (Lorenz and DeWeaver, 2007).

NAO greatly impacts the European climate (Sect. 1). Figure 8 shows the relationships between the changes in NAO amplitude and changes in DJF mean precipitation and surface temperature over northern Europe, central Europe, and the Mediterranean relative to the *piControl*. The change in NAO amplitude is found to have positive correlations with the change in DJF mean precipitation over northern and central Europe (Figure 8a, b) and a negative correlation over the Mediterranean (Figure 8c). There is no clear relationship between the change in NAO amplitude and the temperature change (Figure 8d to f). The *lgm* experiment displays a weakened NAO amplitude along with colder mean temperatures, and the *abrupt4xCO2* experiment shows enhanced NAO amplitude and warmer temperature conversely. The European temperature change overall displays a positive correlation with the change in the NAO amplitude (Figure 8d to f). Both the pattern and amplitude of the NAO weaken in the cold *lgm* experiment and strengthen in the warm *abrupt4xCO2* experiment. It suggests that the NAO is sensitive to change in the mean state. Terrestrial proxy data suggest a warmer winter in Europe during the mid-Holocene (Bartlein et al., 2011) and the last interglacial (Brewer et al., 2008). However, models do not capture this warming in Europe in both experiments and even present a uniform cooling in the *lig127k*. Mauri et al. (2014) suggest that models fail to capture the full extent of high-latitude warming over Northern Europe and underestimate the cold Mediterranean temperatures, because they do not capture a positive NAO-like pattern. This might indicate that the changes in climatological sea level pressure (Fig. 2a) are underestimated in the ensemble.

Earlier studies have noticed a teleconnection existing between ENSO and NAO (Geng et al., 2024; Toniazzo and Scaife, 2006; Hardiman et al., 2019; Zhang et al., 2015; Joshi et al., 2021), though it remains controversial with large uncertainty (López-Parages et al., 2015; Zhang et al., 2019). The ENSO signals reach the North Atlantic region via the stratosphere



(Ineson and Scaife, 2009) and tropospheric pathways (Jiménez-Esteve and Domeisen, 2018). Here we investigate whether a weakened/strengthened ENSO could remotely influence the NAO amplitude. The NAO-ENSO relationship responds nonlinearly to changes in ENSO strength (Hardiman et al., 2019). Enhanced ENSO events affect NAO variability by exciting Rossby waves reaching the North Atlantic through strengthened convection anomalies in the Gulf of Mexico and the Caribbean Sea (Ayarzagüena et al., 2018) and in the tropical Indian and Pacific Ocean (Abid et al., 2021; Joshi et al., 2021). Here, we examine the relationship between change in the strength of ENSO, measured as the standard deviation of nino3.4 index, and change in NAO amplitude (Figure 9). Consistent with Brown et al. (2020), ENSO strengths in the two orbital-forced experiments are weakened (more obviously in the *lig127k* experiment), while they are ambiguous in the *lgm* and *abrupt4xCO2* Figure 9. In contrast, the NAO amplitude weakens in the *lgm* and its changes are ambiguous in the *abrupt4xCO2*, *midHolocene* and *lig127k* experiments (Figure 9). There is no clear relationship between the amplitude changes of ENSO and NAO (Figure 9). It implies that changes in ENSO strength do not remotely influence the NAO amplitude and, therefore, suggest a more nuanced teleconnection then posited by earlier studies.

## 7 Conclusions

We assessed changes in NAO amplitude and teleconnections in a combination of CMIP5 and CMIP6 models. The simulations for past climates include two altered orbital configurations at the mid-Holocene (*midHolocene*) and LIG (*lig127k*) and a colder-than-present LGM (*lgm*), and one for the future as an idealised warming projection of abrupt quadrupling of CO$_2$ (*abrupt4xCO2*). The *lgm* and *abrupt4xCO2* are analysed for NAO response to CO$_2$ forcing. For model evaluation, we compared the *piControl* simulation with the C20 Reanalysis dataset (Compo et al., 2011), as shown in Sect. 3. The *piControl* ensemble reproduces the observed mean state but with biases. The patterns of the positive phase of the NAO and associated teleconnections are consistent between observation and the *piControl* but with variations across the simulations. We explored changes in NAO spatial patterns and explained variance and amplitude. Our results suggest that the NAO change is sensitive to GHG-forcing-induced temperature change but not the orbital configurations. NAO weakens in response to cooling and strengthens to warming. Some of our results are inconsistent with earlier studies, but the inconsistencies are likely affected by the outputs of a few models. The underlying mechanisms are unclear, but previous have highlighted the influence of tropospheric and stratospheric climate changes on NAO behaviour (Rind et al., 2005a, b; Lü et al., 2010). We also discussed changes in NAO teleconnections. The simulated spatial teleconnection patterns associated with the NAO are generally consistent with the theory (see Sect. 1) and are sensitive to the change in NAO amplitude. The *midHolocene* and *lig127k* experiments do not show a clear change in temperature and precipitation associated with the NAO. The weakened NAO in the *lgm* is associated with a cooler and drier northern Europe, while the enhanced NAO in the *abrupt4xCO2* causes a warmer and wetter northern Europe as compared to the *piControl*. We did not find the ENSO-NAO teleconnection suggested by other studies. Further work is required to fully understand the underlying mechanisms driving the NAO change and the associated teleconnections.



390 *Code and data availability.* The processed data for figures are available at https://github.com/annizhao1994/PMIP_NAO and https://doi.org/10.5281/zenodo.15624480.

*Author contributions.* A.Z. and C.B. performed the bulk of the writing and analysis. A.V. put a large effort into the early stage of the analysis. X.S. contributed to the lgm analysis and revised this manuscript. Y.H. revised the manuscript.

*Competing interests.* The authors declare that none of the authors has any competing interests.

395 *Acknowledgements.* We acknowledge the modelling groups that donated their simulation output and the Earth System Grid Federation for distributing all that output. This research has been supported by the National Natural Science Foundation of China (grant no. 42488201). Y.H. is founded in part by the National Natural Science Foundation of China, under grant 42488201. X.S is supported by the Southern Marine Science and Engineering Guangdong Laboratory (Zhuhai) (grant no. SML2023SP204) and the Ocean Negative Carbon Emissions (ONCE) Program.



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





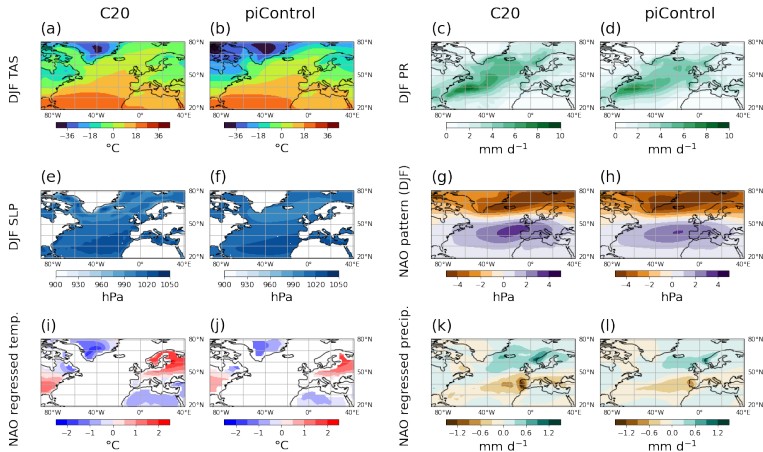

**Figure 1.** Comparison between observations (left column) taken from the 20th-Century Reanalysis (Slivinski et al., 2019) and the ensemble mean of the *piControl* (right column). Panels show DJF mean variables over the North Atlantic sector (see Sect. 2.3 for region definition). The rows present the sea surface temperatures (a, b), precipitation (c, d), sea level pressure (e, f), NAO pattern (g, h), and teleconnections between NAO and temperature (i, j) and precipitation (k, l) computed by regression against the NAO.




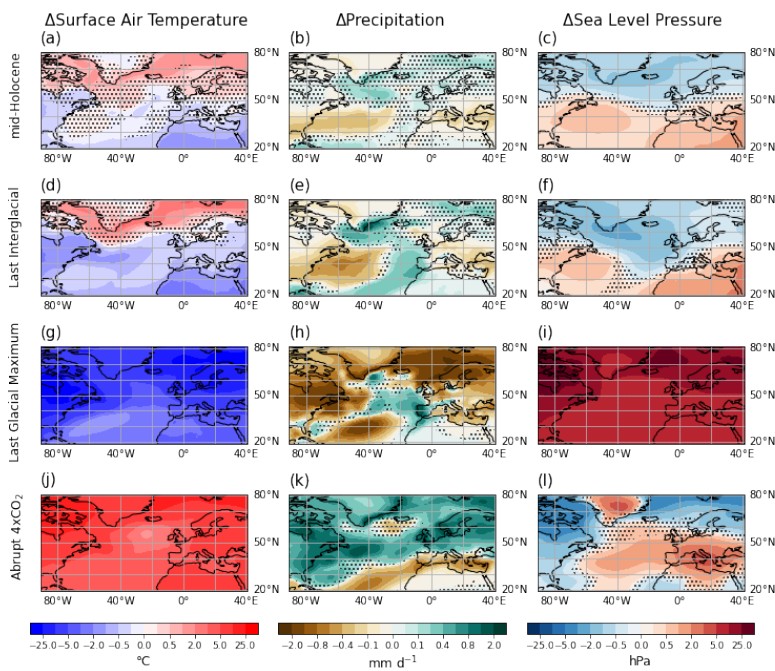

**Figure 2.** Ensemble mean change in mean states. The columns from left to right show the DJF mean surface air temperature (°C; a, d ,g, j), precipitation (mm d$^{-1}$; b, e, h, k) and sea level pressure (hPa; c, f, i l). The rows show the ensemble mean difference from the *piControl* simulations for the *midHolocene* (a, b, c), *lig127k* (d, e, f), *lgm* (g, h, i), and *abrupt4xCO2* simulations (j, k, l). Stippling indicates where the ensemble is inconsistent in the direction of change.

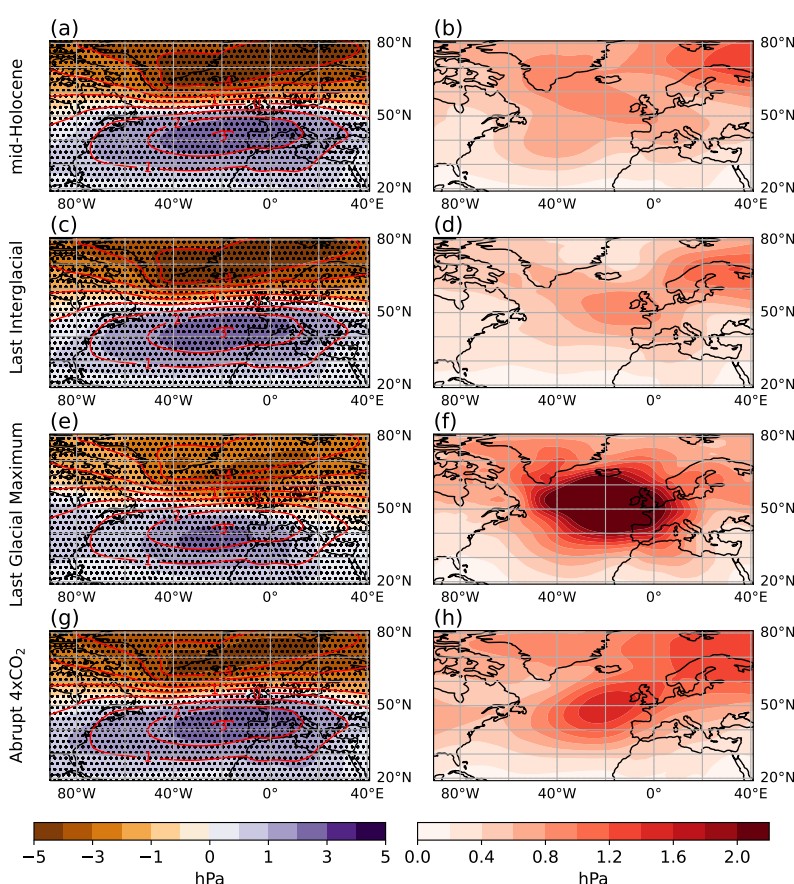

**Figure 3.** SLP spatial pattern for NAO in DJF (hPa) in each experiment, including the *midHolocene* (a), *lig127k* (c), *lgm* (e) and *abrupt4xCO2*(g). Red contours represent the pattern of the *piControl*, and stippling indicates where the ensemble is inconsistent in the direction of change. Ensemble variation is presented as the standard deviation across the ensemble (b, d, f, h).





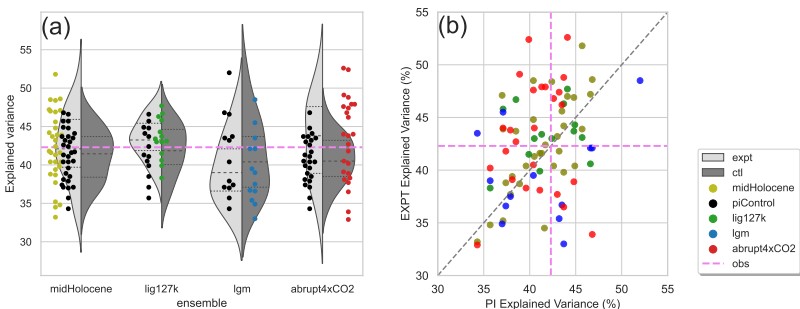

**Figure 4.** Total variance explained by the NAO for (a) the experiments and (b) comparison with the *piControl*. (a) For each column, the dots (corresponding to the left y-axis) show the explained variance by models. The distributions are computed via a kernel density estimation (Waskom, 2021): The curves show the distributions of the explained variance by models for each experiment (left) and the *piControl* (right); horizontal dashed and dotted lines represent the median and 75% and 25% quartiles, respectively. The horizontal location of dots within each column does not have any meaning.





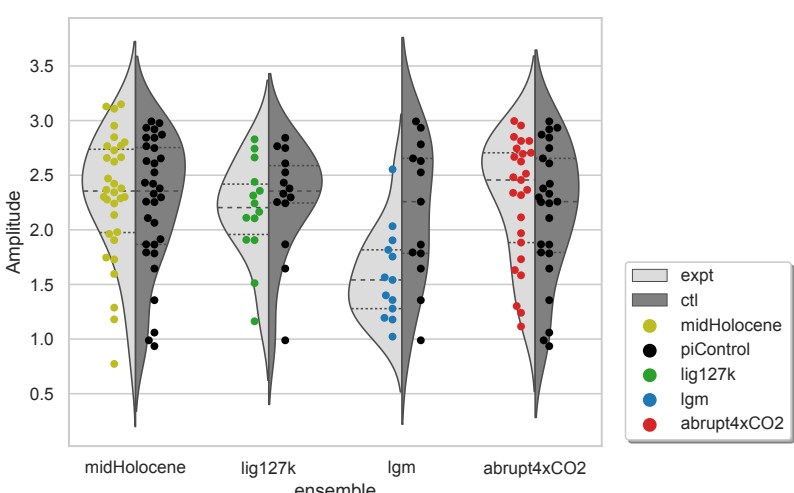

**Figure 5.** The NAO amplitude (hPa) for the experiments. The amplitude is measured as the standard deviation of the renormalised NAO PC time series (Sect. 2.3.).





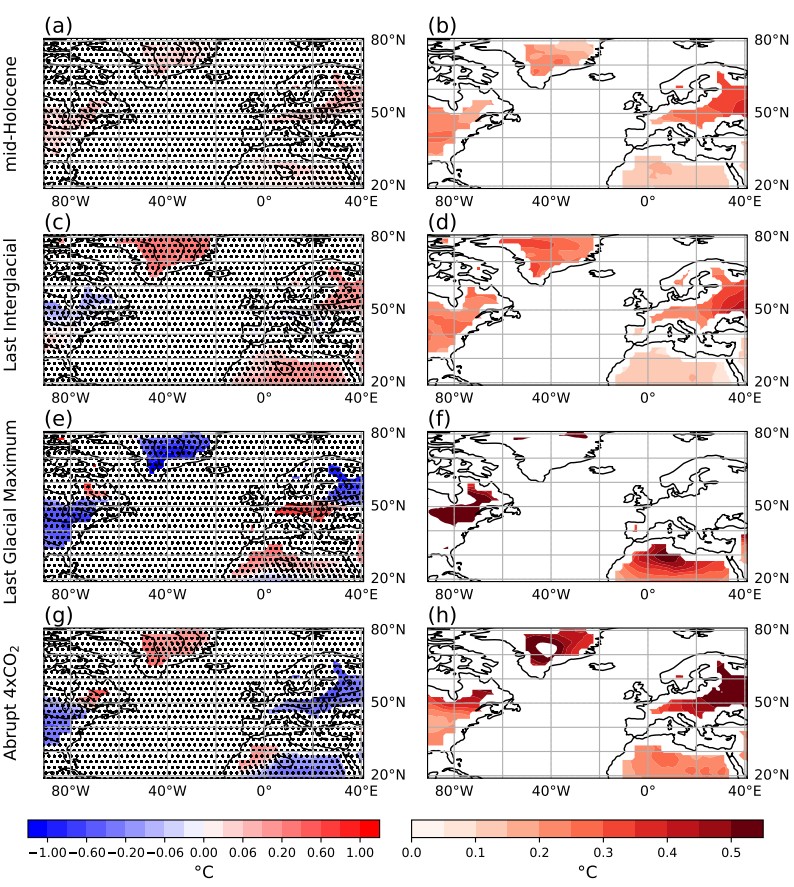

**Figure 6.** Same as Figure 3 but for ensemble mean change in DJF surface mean temperature (°C) associated with the NAO. Stippling indicates where the ensemble is inconsistent in the direction of change.



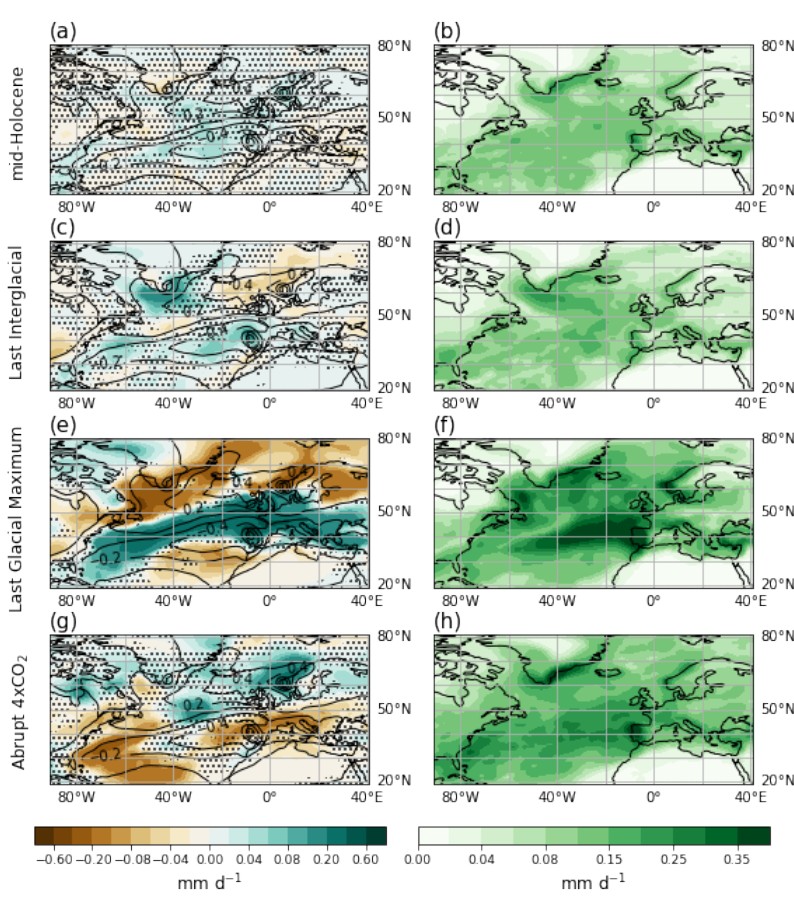

**Figure 7.** Same as Figure 3 but for ensemble mean change in DJF precipitation (mm d$^{-1}$ associated with the NAO. Stippling indicates where the ensemble is inconsistent in the direction of change.





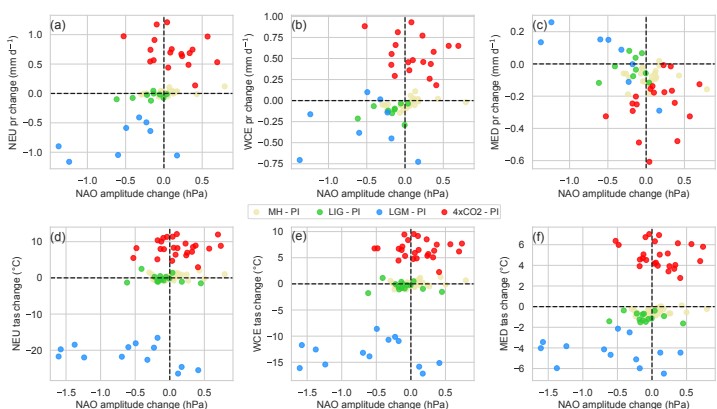

**Figure 8.** Relationships across all experiments and all models between the changes in NAO amplitude and the changes in European mean state of DJF mean precipitation (mm d$^{-1}$; labelled as pr in panels a to c) and surface temperature (°C; labelled as tas in panels d to f). Regions follow the definition of IPCC AR6: northern Europe is referred to as NEU, west and central Europe as WCE and Mediterranean MED. All changes are relative to the *piControl*.





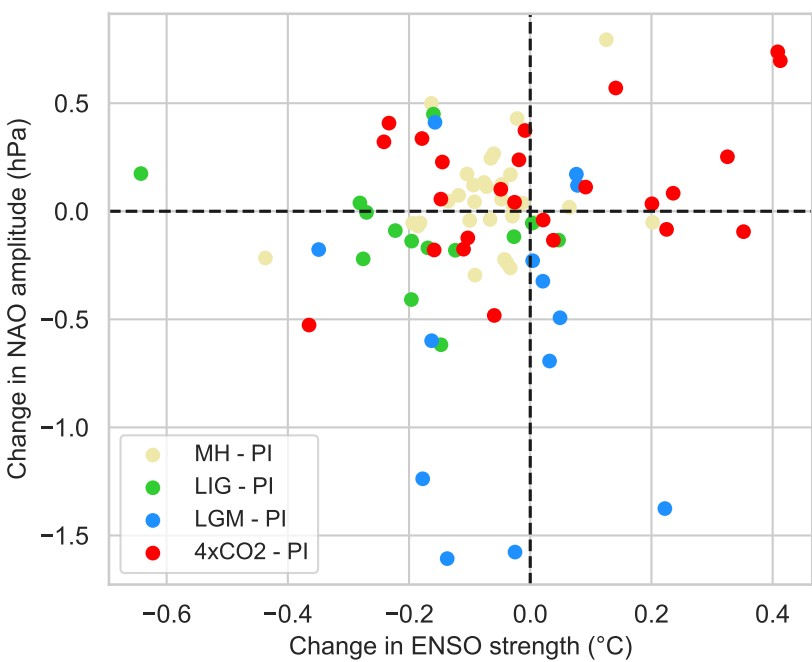

**Figure 9.** Relationship between the change in ENSO strength and NAO amplitude across all models and all experiments relative to the *piControl*. The ENSO strength is measured as the standard deviation of the nino3.4 index of each simulation.



**Table 1.** Models contributing to the control experiment (*piControl*), the PMIP experiments (*midHolocene* (MH); *lig127k* (LIG); *lgm* (LGM)) and an idealised $CO_2$ experiments (*abrupt4xCO2* (4xCO2)) under the CMIP5 and the CMIP6. Numbers in Participation represent the length of model years of each simulation.

| Model | PMIP | ECS* (K) | Participation PI | MH | LIG | LGM | abrupt4xCO2 | REF. |
|---|---|---|---|---|---|---|---|---|
| AWI-ESM-1-1-LR | 4 | 3.6 | 100 | 100 | 100 | 100 | | Sidorenko et al. (2015) |
| BCC-CSM1-1 | 3 | 3.1 | 500 | 100 | | | 50 | Xin et al. (2013) |
| CCSM4 | 3 | 2.9 | 1051 | 301 | | 101 | 50 | Gent et al. (2011) |
| CESM2 | 4 | 5.3 | 500 | 700 | | | 50 | Gettelman et al. (2019) |
| CNRM-CM5 | 3 | 3.3 | 850 | 200 | | 200 | 50 | Voldoire et al. (2013) |
| CNRM-CM6-1 | 4 | 5.1 | 500 | | 301 | | 50 | Craig et al. (2017) |
| COSMOS-ASO | 3 | | 399 | | | 600 | | Wetzel et al. (2010) |
| CSIRO-Mk3-6-0 | 3 | 4.1 | 500 | 100 | | | 50 | Rotstayn et al. (2011) |
| CSIRO-Mk3L-1-2 | 3 | 3.1 | 1000 | 500 | | | | Phipps et al. (2012) |
| EC-EARTH-2-2 | 3 | 4.2 | 40 | 40 | | | | Hazeleger et al. (2012) |
| EC-Earth3-LR | 4 | 4.3 | 201 | 203 | 210 | | | Döscher et al. (2021) |
| FGOALS-f3-L | 4 | 3 | 561 | 500 | 500 | | 50 | He et al. (2020) |
| FGOALS-g2 | 3 | 3.7 | 700 | 680 | | 100 | 50 | Li et al. (2013) |
| FGOALS-g3 | 4 | 2.9 | 700 | 500 | 500 | | | Li et al. (2020) |
| FGOALS-s2 | 3 | 4.5 | 501 | 100 | | | 50 | Bao et al. (2013) |
| GISS-E2-1-G | 4 | 2.7 | 851 | 100 | 100 | | 50 | Kelley et al. (2020) |
| GISS-E2-R | 3 | 2.1 | 500 | 100 | | 100 | 50 | Schmidt et al. (2014) |
| HadGEM2-CC | 3 | 4.5 | 240 | 35 | | | | Collins et al. (2011) |
| HadGEM2-ES | 3 | 4.6 | 336 | 101 | | | 50 | Collins et al. (2011) |
| HadGEM3-GC31-LL | 4 | 5.4 | 100 | 100 | 200 | | | Williams et al. (2018) |
| INM-CM4-8 | 4 | 2.1 | 531 | 200 | 100 | 200 | 50 | Volodin et al. (2018) |
| IPSL-CM5A-LR | 3 | 4.1 | 1000 | 500 | | 200 | 50 | Dufresne et al. (2013) |
| IPSL-CM6A-LR | 4 | 4.5 | 1200 | 550 | 550 | | 50 | Boucher et al. (2020) |
| KCM1-2-2 | 3 | | 200 | 100 | | | | Park et al. (2009) |
| MIROC-ES2L | 4 | 2.7 | 500 | 100 | 100 | 100 | 50 | Hajima et al. (2020) |
| MIROC-ESM | 3 | 4.7 | 630 | 100 | | 100 | 50 | Sueyoshi et al. (2013) |
| MPI-ESM-P | 3 | 3.5 | 1156 | 100 | | 100 | 50 | Giorgetta et al. (2013) |
| MPI-ESM1-2-LR | 4 | 2.8 | 1000 | 500 | 300 | | 50 | Mauritsen et al. (2019) |
| MRI-CGCM3 | 3 | 2.6 | 500 | 100 | | 100 | 50 | Yukimoto et al. (2012) |
| MRI-ESM2-0 | 4 | 3.1 | 701 | 200 | | | 50 | Yukimoto et al. (2019) |
| NESM3 | 4 | 3.7 | 100 | 100 | 100 | | 50 | Cao et al. (2018) |
| NorESM1-F | 4 | 2.3 | 200 | 200 | 200 | | | Guo et al. (2019) |
| NorESM2-LM | 4 | 2.5 | 391 | 100 | 100 | | 50 | Seland et al. (2020) |
| UofT-CCSM-4 | 4 | 3.2 | 100 | 100 | | 100 | | Chandan and Peltier (2017) |
| Ensemble size | | | 34 | 32 | 14 | 13 | 23 | |