# Peer review of "North Atlantic Oscillation (NAO) in the Paleoclimate Modelling Intercomparison Project (PMIP)"

_EGUsphere, 2025_

## Author Comment (AC2)

**Reply to Referee 3: egusphere-2025-3140**

A. Zhao *et al.*

**Correspondence:** yyhu@pku.edu.cn

We thank the reviewer for their diligence in reviewing our manuscript. We already aim to revise the text to incorporate better use of topic sentences (as per Reviewer 2) and will incorporate improvements to the readability and clarity feedback of the text in light of this work. Specifically, we will restructure the introduction to mechanisms of the NAO, and we will avoid referring to the LGM as a predominantly GHG-driven change whilst further emphasising the orographic steering effects. Additionally,
5 we will tighten the references and address the presentational issues. Blue text below is our response to the referee's comments (reproduced in black).

**Referee 3**

Review of the manuscript „North Atlantic Oscillation (NAO) in the Paleoclimate Modelling Intercomparison Project (PMIP)" by Zhao et al.

10 Overview

Zhao et al. present an analysis of the North Atlantic Oscillation (NAO) from results of simulations of multiple iterations of the Paleoclimate Modelling Intercomparison Project (PMIP) and the Climate Modelling Intercomparison Project (CMIP). To study the impact of background climate on NAO and the climate of the North Atlantic realm, the authors employ model results from abrupt-4xCO2, a CMIP DECK simulation that aims at sampling model sensitivity to a very strong carbon dioxide
15 forcing (Eyring et al., 2016; Dunne et al., 2025), the CMIP/PMIP piControl as reference, two mainly orbitally controlled PMIP simulations lig127k and midHolocene, and the Last Glacial Maximum simulation lgm that is forced by a combination of orbit, greenhouse gas concentrations, and geography. Simulations lgm and abrupt-4xCO2 represent cold and warm end members. This allows to sample the response in NAO to climate backgrounds that differ strongly from the current climate, to reflect on changes in Europe's climate as a result from such extreme climate backgrounds, and to compare model results with the
20 theoretical framework provided by previous work.

In their introduction the authors provide an overview on NAO mechanisms and of first order impacts on climate, which provides a good stage for linking this study with previous work on the NAO. Analysis of model output is performed based on a pre-existing framework by Brierley and Wainer (2018) and follows earlier work by Zhao et al. (2022). On the side of results, the authors find that models can reproduce in their piControl simulation inferences derived from 20th century Reanalysis (Compo et
25 al., 2011), with the exception of certain temperature, sea level pressure, and precipitation biases. From the climate perturbation simulations lgm, abrupt-4xCO2, midHolocene and lig127k it becomes clear that first-order impacts of climate on NAO and European climate are confirmed, in particular dependency of the expression of NAO on the mean state. Nevertheless, there are

also more nuanced results. In particular, a previously established relationship between NAO and El Nino Southern Oscillation is not confirmed by this study. Models have difficulty to reproduce the warmer winter climate reconstructed for Europe for middle Holocene and Last Interglacial. Generally, there is substantial model-dependency on results. I find this aspect particularly well illustrated by the differences in NAO metrics that are derived for differently composed model ensembles of the piControl simulation presented in Figures 4 and 5.

I have read the manuscript with interest and I find it relevant for publication in Climate of the Past. It is obvious that the authors have invested substantial work in aggregating a very large ensemble of different models across different phases of CMIP and PMIP towards enabling a broad analysis and deliver a manuscript that may be of value to a wide readership in climate sciences. This is certainly laudable. On the other hand I think that various aspects of the manuscript deserve improvement, also towards enabling the readership to more easily follow the presentation and interpretation of the results provided by the authors. I suggest reconsideration after major revisions. While many of my comments address minor issues, sometimes even only technicalities, in some aspects I was left with further questions that are in my humble opinion best solved in a second iteration of the manuscript. Please find more detailed information on suggestions for improvements below.

We would like to thank you for the constructive and detailed comments on our manuscript. We would be revise the work as suggested, as we agree that it would improve the manuscript's readability and therefore impact.

Overarching comments

At several locations, authors' description of results was difficult to follow. The manuscript could be improved by using more links that refer from text to figures. I think that particularly Section 5 could be improved in this respect. From the perspective of a non-native speaker I also noticed at several locations unclear or ambiguous formulations that sometimes obstructed my understanding of the authors' statements. Generally, the manuscript could be improved by fixing various typos and formulations, by making sure that referencing of previous work is at some locations more precise (particularly for model description papers), and by ensuring that definitions and descriptions of figure content are more comprehensive by adding detail to figure captions. Please also check capitalization rules (e.g. „Last Glacial Maximum" vs. „the last interglacial", both used in this manuscript). I note that due to the figure size in the typeset PDF, that is for the majority of figures far too small, I found it cumbersome to interpret various results. Please check formatting and content of your reference list, e.g. https://doi.org/https://doi.org/10.1029/2000GL012377.

Thanks for the suggestion. We will enhance the linkages between text and figures. We will also improve the manuscript quality, including language, figures and references.

Generally, I think that a bit more consistency across the manuscript when attributing changes in NAO to different types of boundary conditions and forcings used in the various simulations would be beneficial towards clarity. Discussion of the impact of geography (in particular ice sheet height and extent) should be extended. Generally, I would avoid reducing lgm to its greenhouse gas forcing (this is done at various locations of the manuscript, e.g. „Our results show that the NAO is sensitive to GHG-forcing-induced temperature changes but not the orbital configurations", while the authors acknowledge at other locations that reducing lgm to its greenhouse gas forcing alone is an oversimplification, e.g. line 191). Even if analyses of the relative contribution of different forcings and boundary conditions to the overall climate patterns and expression of NAO

are not possible due to a lack of availability of higher Tier simulations proposed by Kageyama et al. (2017), there is literature around (e.g. Justino and Peltier, 2005) based on which expected changes could be discussed and promoted for more detailed analysis in future work. For example, Pausata et al. (2011) show that details of topography are key players in controlling atmosphere circulation and patterns, of which many are key to the NAO.

*This framing had been inherited from our previous work on tropical climate variability – where the ice-sheets do have a relatively minor impact. We acknowledge that the North American ice-sheets clearly play a role in steering the dynamics of the North Atlantic storm-track. We will revise the text.*

When transferring knowledge of the quite idealized abrupt-4xCO2 simulation to real world examples of future climate one could increase reflection of model biases in the simulated strength of NAO (Outten and Davy, 2024) and weakened NAO variability in future scenarios (Fuentes-Franco et al., 2023).

*We will correct the terminology for the abrupt-4xCO2 simulations.*

While I generally like the introduction of the mechanisms behind NAO, I think that explanation of the mechanisms behind NAO and the effects on regional climate of (different phases of) the NAO (first two paragraphs of the introduction) could be improved to be easier to follow. I think that it would be clearer if the explanation related to the pressure systems (lines 16-17, lines 33-35, lines 37-39, lines 44-45) were grouped together, and description of the various effects of NAO on climate would then follow thereafter. That would also increase clarity regarding the two „pressure systems" that you talk of in line 33/34 and their link to the sea level pressure anomalies (line 16) via which the NAO is commonly defined.

*Thanks! We will revise the Introduction.*

As noted above, there are various inconsistencies in referencing. Please check all references carefully, there may be more issues than I have been able to locate. Please pay particular attention to Table 1 (see also a comment further down) and to references to the 20th Century Reanalysis, that you appear to cite as Slivinski et al. (2019) in the caption of Figure 1 but as Compo et al. (2011) in line 166/167.

*We will fix the inconsistencies in referencing.*

Specific comments:

line 2-3: „whose future behaviour remains uncertain" - I note that this statement could be attributed to any of the three statements made in the sentence, and it is not clear to me which one the authors would like to single out here (if any).

*It refers to the behaviour of NAO in the future. We will clarify it during revision.*

line 6: I think that the terms „future warming scenario" or „climate scenario" refer to specific trajectories of human-made emissions / greenhouse gas concentrations. See e.g. the statement by Dunne et al. (2025): „CMIP provision of climate responses to idealized and scenario-based projections of forcing has supported numerous national and international assessments [...] and played a central role in every Intergovernmental Panel on Climate Change (IPCC) report since its inception [...]. Scenario projections include the response to changes in CO2 and other greenhouse gases, aerosols, and ozone across a range of increasing and recovery trajectories via human perturbations to the carbon cycle and other aspects of the Earth system". I think the common terminology for abrupt-4xCO2 is that of a baseline simulation from the CMIP DECK or similar (Eyring et al., 2016).

*We will change it to "idealized GHG-forced experiment from the CMIP DECK".*

line 8: „inducing variations in the seasonal cycle" - and redistributing energy across latitudes

We will modify it during revision.

line 8: lgm is characterized also by major differences in ice sheets (albedo and elevation), that are known to impact atmospheric variability („This demonstrates that the topographic forcing and ice albedo feedback are both crucial to generating the quadrapolar form of the atmospheric variability, primarily due to the attendant modifications of the stationary and transient waves." (Justino and Peltier, 2005)). This could be pointed out towards not underselling your work, and I think the reference could also be included in your introduction and discussion regarding previous work.

We will revise the text during revision.

line 18: I think the work by Hurrell et al. (2003) that you cite here to highlight importance of the NAO in the North Atlantic realm provides actually a more nuanced view: „[...] although the NAO is the dominant pattern of atmospheric circulation variability over the North Atlantic, it explains only a fraction of the total variance, and most winters cannot be characterized by the canonical NAO pattern"; this may be something to reflect on in your discussion.

Thanks for the suggestion. We will modify it during revision.

line 43: „Research around changes in the NAO involves some awkward terminology." Towards increased understanding on the side of your readership it would be illustrative if you provided details on specifics of the common terminology that you consider awkward. From the information that follows one could infer various different characteristics as awkward, including the definition of NAO via pressure differences between two pressure systems who's extent and location changes with the background climate (your lines 57ff).

We will add examples here for explanations.

line 56: replace „as" by „having been linked with the attributes"? For clarity I also suggest to start a new sentence after the reference to Gladstone et al. (2005).

We will revise the text as suggested to improve clarity.

line 62: „shifts positively" - do you mean that the NAO shifts towards being more predominantly in its positive phase? Please clarify formulation.

Yes. We will clarify it.

line 70: see above comment regarding differences between the common terminology of scenario vs. abrupt-4xCO2.

We will revise the terminology.

line 72: „warming projection" as per my previous comment

line 80: consider replacing colloquial language („a bunch of simulations") by more formal formulation, e.g. „a group of simulations that spans across ..."

We will replace the text as suggested.

line 110: remove the comma for clarity?

Thanks for the suggestion. We will remove the comma.

line 117: add space after the word „reconstructions"

We will add space.

line 142: Question about the state of equilibrium between the different experiments and simulations by individual modelling groups: „we take the last 50 model years of the abrupt4xCO2 simulations in analysis" - here it would be illustrative if you added information on the length of the simulation as defined in a) CMIP and as b) performed by individual modeling groups to the manuscript. b) would be relevant if modeling groups for some reasons have performed longer simulations than the suggested standard length. This comment actually also is relevant for all other simulations that you present.

Thanks for the suggestion. We will add a brief description here. The CMIP protocol specifies a minimum length of 150 years for the abrupt4xCO2 simulations to reach near-equilibrium (Eyring et al., 2016). We take the last 50 years to ensure results reflect near-equilibrium conditions, facilitating direct comparison with other equilibrium experiments in our study. For the piControl experiment, CMIP6 requires a minimum of 500 years to ensure a equilibrium state (Eyring et al., 2016). The PMIP simulations do not have a fixed length. The length depends on when the simulations meet the criteria: (a) the absolute trend in global mean sea surface temperature is less than 0.05 K/century, and (b) the AMOC is stable (Kageyama et al., 2018).

line 147: replace „as" with „via the threshold that"?

We will replace the text as suggested.

line 151: delete „-based" from the definition of the abbreviation for clarity - later on you refer just to PC.

We will delete it.

line 167: „show bias" „are biased" instead of „but bias"?

We will revise it.

line 176: This long sentence is difficult to grasp - try to split it, e.g. after the comma in line 176, and at the junction between agreement and disagreement between models and observations.

We will split the sentence.

line 179: „the NAO explains 41.5% (34.3%-52.0%) of the total variance" . Please outline how this statement is derived, I assume you could refer here to Fig. 4, potentially to the pink dashed lines (but I am not sure)?

It relies on the dots. We will clarify it.

line 183: fix brackets of reference for clarity

We will revise it.

line 231: not sure what the authors mean with the formulation „AMOC condenses"

We will revise and clarify the terminology.

line 246ff: For the whole section, please add more references to analyses/figures from which you derive your statements. This is not always obvious. It would be good if it was made possible to the reader to identify from the presented results (in particular figures) the models that behave differently than others. Not sure whether another table could help here. Adding model names or index numbers to data points would probably lead to too busy figures?

Sorry for the inconvenience. We will add links to figures. For the possibility of adding an extra table here, we feel it might not be necessary, as we are focusing more on the experiments as a whole rather than evaluating individual model performance.

line 259: the meaning of the sentence is unclear to me. Do you mean „The absence of clear changes in explained variance between midHolocene and piControl is consistent with Gladstone et al. (2005)"?

We will revise the text to clarify the consistency.

170 line 272: „the NAO weakens than the piControl"? please check formulation

We will change to "the NAO weakens compared to the piControl".

line 278: add space in bracket

We will add the space.

line 281/283: largest „reduction" instead of „weakness"?

175 We will revise it.

line 285/286: spacing in bracket

We will revise it.

line 290: What does „it" refer to here? Please clarify text accordingly.

"it" refers to the averaged NAO amplitude in the abrupt4xCO2 simulations. We will clarify it during revision.

180 line 293: add closing bracket

We will add it.

line 300/304: „AO" refers here to Arctic Oscillation, it is not a typo, right? Please define the abbreviation here. This is particularly important since in line 20 you use a completely different abbreviation for Arctic Oscillation.

AO here refers to Arctic Oscillation. We will change it to "Arctic Oscillation".

185 line 306: positively -> increases; negatively -> reduces? Maybe one can find better terms for warming and cooling to highlight focus on the mean climate rather than on a transient (maybe warmer or colder background states?)

We will change the words.

line 314: cooler conditions? If „cooling" is used the sentence should be fixed for grammar.

We will change it to "cooler conditions".

190 line 316: while it decreases?

We will revise it.

line 318: add missing dot

We will add it.

line 327: weakness -> reduction of the importance of?

195 We will revise it.

line 332: „than" -> compared to

We will revise it.

line 345ff: „There is no clear relationship between the change in NAO amplitude and the temperature change (Figure 8d to f) [...] The European temperature change overall displays a positive correlation with the change in the NAO amplitude (Figure

200 8d to f)". Not sure what I miss, but here you lost me.

Sorry about that. We will clarify it during revision.

line 366: add brackets around Figure 9

We will revise it.

line 370: than

We will revise it.

Line 375: „The lgm and abrupt4xCO2 are analysed for NAO response to CO2 forcing" - to separate CO2 forcing impact from that of other drivers one would need a forcing factorization, which is not made here. More precise formulation would be to state that: „The lgm and abrupt4xCO2 are analysed, attributing NAO response to CO2 forcing." Please refer to my reservations noted above to reduce the LGM forcing to just the CO2 as a driver. LGM is the most complex situation among the simulations considered here, combining ghg forcing, orbital forcing, and changes in prescribed paleo-geography. Furthermore, one could more precisely formulate the sentence thereafter: „For evaluation of models for the reference climate state ..."

We will revise the text to avoid reducing LGM to CO2.

line 380: „GHG-forcing-induced temperature change but not the orbital configurations" - see above

line 380/381: „NAO weakens in response to cooling and strengthens to warming" - based on results shown in your manuscript it could also be that the absence of sensitivity to orbital forcing is just a result of the much reduced global change in background climate in midHolocene and lig127k if compared to abrupt-4xCO2 and lgm, right? So one could formulate lines 385 a bit more explicitly in this regard?

We will revise it.

line 381: „Some of our results are inconsistent with earlier studies, but the inconsistencies are likely affected by the outputs of a few models." I think model-dependency is quantitatively shown by variation of results on NAO metrics in the different piControl ensembles, Figs. 4/5. This could be highlighted.

We will highlight model spread here.

line 382: previous studies

We will revise it.

Figures: Generally Figures are typeset too small. I had to zoom into the PDF to 400% to be able to identify details. I think that there is enough space to increase the figures to linewidth so that they are more easily to study.

Thanks for the constructive comments on figures! We will enlarge the figures and improve their quality as suggested below.

Figure 1: in a,b you show surface air temperatures, not sea surface temperatures as stated; please specify which temperatures you show in (i,j)

We will clarify variables.

Figure 3: add space after abrupt4xCO2

We will add space.

Figure 4: „The horizontal location of dots within each column does not have any meaning." - change to „In a), horizontal location of dots within each column has been offset for better visibility"? - nevertheless, I am asking myself why the offset changes between experiments, where point clouds in abrupt4xCO2, for example, are much more clearly separated from each

other than for midHolocene; please state the difference in black and pink lines - I assume the latter refer to the PI NAO variance explained?

We will revise the figure.

Figure 4a and 5: so "ctl" in the legend stands for piControl, and "expt" stands for the respective experiment shown as a label on the x-axis? Please add such definitions to the figure caption. Furthermore, if this statement is correct, then it is a bit confusing that the black piControl dots are not located within the darker gray shade of ctl; similar for expt. From the fact that the shape of ctl differs a lot between experiments I deduce that the NAO variance explained by the piControl state strongly depends on the selection of models that are considered for computation of the piControl metrics. This could be reflected on in the discussion regarding uncertainties in the change for different experiments.

We will revise the figure caption.

Figure 6: please be specific in the surface mean temperature that you use here. Is it the same as in Figure 2, where you state that you use surface air temperature?

We will revise it.

Figure 7: closing bracket missing at precipitation unit; continental outlines on the left hand side difficult to decipher among the stippling - use different colors?

We will fix the unit formatting and improve the figure.

Figure 8: surface air temperature?

We will clarify it.

Table 1: It is laudable that the authors invested substantial effort to collect information for a comprehensive overview based on which readers can access more information on the employed models. On the other hand, it appears that some things went wrong here. I am by no means an expert in each and every model that is presented here, but several of the references leave me puzzled. At least one reference is not listed in the reference list of the manuscript (Wetzel et al., 2010) and, although being relatively familiar with the model family, I have not been able to locate the respective publication. The only references that may come close are Budich et al. (2010) (gray literature) and Jungclaus et al. (2006) or Jungclaus et al. (2010). Which one fits better really depends on the context that the authors have in mind. At least one reference does not seem to fit to the respective model at all: Craig et al. (2017) do not seem to describe the model CNRM-CM6-1 as advertised here but rather present development and performance of the OASIS3-MCT_3.0 toolbox that is used by various coupled climate models. While one reference (Chandan and Peltier, 2017) refers to the University of Toronto version of CCSM4 as advertised, the manuscript describes the mid-Pliocene experiment of PlioMIP2 and seems (I may be wrong here) a very specific development for PlioMIP simulations which are in fact not part of the current study. Is this manuscript still a meaningful reference for this model in the context of your publication or are different manuscripts more relevant to cite here? I speculate that Chandan and Peltier (2020) may be relevant for the mid-Holocene, but again, I am no expert in all the models.

The U-of-T variant of CCSM4 is that used by Chandan and Peltier (2017), but was originally modified by Vettoretti and Peltier (2014). Nonetheless, the reviewer is correct that Chandan and Peltier (2020) is probably more appropriate in this context.

270     We have processed the NAO for all the Pliocene simulations, but chose not to include them as the scope of the paper was already a little broad.

    Supplement:

    „The web address can be created manually by adding https://doi.org/ in front of each doi." If one would like to access a larger amount of files via your, indeed very nice, list, then this would entail quite some typing and copying work. Why not
275 directly provide the web addresses as clickable download links here?

    Thanks for the suggestion. We will directly provide the web addresses instead.

    „Those simulations that are not the ESGF (no available doi) were required from the model group, and are marked as N/A here instead." Please check the sentence. Should it read „Data of those simulations, that are not available via the ESGF (no available doi), were obtained directly from the respective modelling group, and are marked as N/A here instead."
280     We will revise it.

    Also here please check your references. At least two miss some information or have awkward formatting (the two by Seland et al.).

    We will check and revise them during revision.

    Additional References

    Budich, R., Giorgetta, M., Jungclaus, J., Redler, R., and Reick, C.: The MPI-M Millennium Earth System Model: An Assembling Guide for the COSMOS Configuration, https://pure.mpg.de/rest/items/item$_2$193290/$component/file_2$193291/$content$, 2010.
285     Chandan, D., Peltier, W. R.(2020). African Humid Period precipitation sustained by robust vegetation, soil, and lake feedbacks. Geophysical Research Letters, 47, e2020GL088728. https://doi.org/10.1029/2020GL088728.

    Dunne, J. P., Hewitt, H. T., Arblaster, J. M., Bonou, F., Boucher, O., Cavazos, T., Dingley, B., Durack, P. J., Hassler, B., Juckes, M., Miyakawa, T., Mizielinski, M., Naik, V., Nicholls, Z., O'Rourke, E., Pincus, R., Sanderson, B. M., Simpson, I. R., and Taylor, K. E.: An evolving Coupled Model Intercomparison Project phase 7 (CMIP7) and Fast Track in support of future
290 climate assessment, Geosci. Model Dev., 18, 6671–6700, https://doi.org/10.5194/gmd-18-6671-2025, 2025.

    Eyring, V., Bony, S., Meehl, G. A., Senior, C. A., Stevens, B., Stouffer, R. J., and Taylor, K. E.: Overview of the Coupled Model Intercomparison Project Phase 6 (CMIP6) experimental design and organization, Geosci. Model Dev., 9, 1937–1958, https://doi.org/10.5194/gmd-9-1937-2016, 2016.

    Fuentes-Franco, R., Docquier, D., Koenigk, T. et al. Winter heavy precipitation events over Northern Europe modulated by a
295 weaker NAO variability by the end of the 21st century. npj Clim Atmos Sci 6, 72 (2023). https://doi.org/10.1038/s41612-023-00396-1

    Jungclaus, J. H., and Coauthors, 2006: Ocean Circulation and Tropical Variability in the Coupled Model ECHAM5/MPI-OM. J. Climate, 19, 3952–3972, https://doi.org/10.1175/JCLI3827.1.

    Jungclaus, J. H., Lorenz, S. J., Timmreck, C., Reick, C. H., Brovkin, V., Six, K., Segschneider, J., Giorgetta, M. A., Crowley,
300 T. J., Pongratz, J., Krivova, N. A., Vieira, L. E., Solanki, S. K., Klocke, D., Botzet, M., Esch, M., Gayler, V., Haak, H., Raddatz, T. J., Roeckner, E., Schnur, R., Widmann, H., Claussen, M., Stevens, B., and Marotzke, J.: Climate and carbon-cycle variability over the last millennium, Clim. Past, 6, 723–737, https://doi.org/10.5194/cp-6-723-2010, 2010.

Justino, F., and W. R. Peltier (2005), The glacial North Atlantic Oscillation, Geophys. Res. Lett., 32, L21803, doi:10.1029/2005GL023822

Kageyama, M., Albani, S., Braconnot, P., Harrison, S. P., Hopcroft, P. O., Ivanovic, R. F., Lambert, F., Marti, O., Peltier,
305   W. R., Peterschmitt, J.-Y., Roche, D. M., Tarasov, L., Zhang, X., Brady, E. C., Haywood, A. M., LeGrande, A. N., Lunt, D.
J., Mahowald, N. M., Mikolajewicz, U., Nisancioglu, K. H., Otto-Bliesner, B. L., Renssen, H., Tomas, R. A., Zhang, Q., Abe-
Ouchi, A., Bartlein, P. J., Cao, J., Li, Q., Lohmann, G., Ohgaito, R., Shi, X., Volodin, E., Yoshida, K., Zhang, X., and Zheng, W.:
The PMIP4 contribution to CMIP6 – Part 4: Scientific objectives and experimental design of the PMIP4-CMIP6 Last Glacial
Maximum experiments and PMIP4 sensitivity experiments, Geosci. Model Dev., 10, 4035–4055, https://doi.org/10.5194/gmd-
310   10-4035-2017, 2017.

Outten, S. and Davy, R.: Changes in the North Atlantic Oscillation over the 20th century, Weather Clim. Dynam., 5, 753–762,
https://doi.org/10.5194/wcd-5-753-2024, 2024.

Pausata, F. S. R., Li, C., Wettstein, J. J., Kageyama, M., and Nisancioglu, K. H.: The key role of topography in altering North
Atlantic atmospheric circulation during the last glacial period, Clim. Past, 7, 1089–1101, https://doi.org/10.5194/cp-7-1089-
315   2011, 2011.

---

## Author Comment (AC3)

**Reply to Referee 2: egusphere-2025-3140**

A. Zhao *et al.*

**Correspondence:** yyhu@pku.edu.cn

We would like to thank the reviewer for reviewing our manuscript and providing constructive comments. We already aim to revise the text and improve the readability and clarity, as suggested by reviewers. Specifically, we will restructure the introduction to mechanisms of the NAO, and we will avoid referring to the LGM as a predominantly GHG-driven change whilst further emphasising the orographic steering effects. Additionally, we will use better topic sentences as suggested. We will also tighten the references and address the presentational issues. Blue text below is our response to the referee's comments (reproduced in black).

**Referee 2**

Review of "North Atlantic Oscillation (NAO) in the Paleoclimate Modelling Intercomparison Project (PMIP)" by Zhao, et al.

The authors investigate how the mean state of the European climate and the NAO change in three simulated past climate and one idealised future warming climate. Their results highlight that the NAO is sensitive to GHG-forcing-induced temperature change but not the orbital configurations. They also show consistent changes between the amplitude of the NAO and the precipitation.

A key strength of this paper is the comparison of NAO responses to two distinct types of forcings. The results carry important implications for understanding the NAO response to global warming

However, several issues need clarification. In particular, some statements differ from previous studies without sufficient discussion, there are mismatches between text and figures, and the organization of paragraphs could be improved.

Massive thanks. We are happy to make the revisions suggested.

Major points:

Line 64-68: The mean state change in the NAO index shows spreads among models. However, the reason summarized in the text is not correct, because most models predict a reduced temperature gradient at lower-level as well as an enhanced temperature gradient at upper-level (see Fig. 3 Harvey et al, 2015). Actually, as shown by McKenna et al, 2021, this "large spread" is mainly due to internal variability.

- McKenna, C. M., & Maycock, A. C. (2021). Sources of uncertainty in Multimodel Large Ensemble projections of the winter North Atlantic Oscillation. Geophysical Research Letters, 48, e2021GL093258. https://doi.org/10.1029/2021GL093258

We will also add in a discussion of internal variability. Thankfully internal variability is less of a factor in the present work, as we can average over substantially longer periods. We were not aware of McKenna & Maycock (2021), so thank you for

bringing it to our attention. It would appear to suggest that internal variability explains only 1/3 of the spread, rather than being the main source of it.

30

Line 43-48: Changes in the NAO index can be partitioned into changes in its mean state (shift in the NAO index distribution) and changes in its variability (changes in the shape of the distribution), see Liu (2025) and O'Brien and Deser (2023). The last sentence in this paragraph is unclear.Why are these two aspects difficult to separate in paleoclimate reconstructions but "easy to distinguish" in paleoclimate simulations? Further clarification is needed.

35    – Liu, Quan, et al. "More extreme summertime North Atlantic Oscillation under climate change." Communications Earth & Environment 6.1 (2025): 474.

      –

      – O'Brien, J. P. and C. Deser, 2023: Quantifying and understanding forced changes to unforced modes of atmospheric circulation variability over the North Pacific in a coupled model large ensemble. J. Climate, 36, 17-35, doi: 10.1175/JCLI-
40       D-22-0101.1.

Fundamentally, palaeoclimate records consist of individual timeseries at relatively low temporal resolution (annual at best) and often uncertain spacing between the data-points. This alone makes the decomposition between changes at two different timescales challenging. The direct chronological connection between of records at different locations with an accuracy sufficient to be sure they represent the same individual season is not possible. This means that spatial patterns of inter-seasonal
45   variability is out-of-reach.

Line 196 / Line 215 consistency: The midHolocene and lig127k experiments show weaker meridional temperature gradients than the PiControl. According to Line 64, this should lead to a negative NAO-like mean-state change. Yet Line 215 states that the mean state exhibits a positive NAO-like pattern. Do you have an explanation?

50   We apologize for this inconsistency, which may have arisen from the insufficient distinction between the concept of changes in the NAO. The NAO index in future simulations (Lines 61-68) is defined as the difference in DJF zonal mean sea level pressure at fixed latitudes, and "positive/negative" refers to the signal of anomalies considered as the difference in index between the averages from the end of the 21st century under SSP scenarios and averages from 1995–2014. In our study, Line 215 describes the change in sea level pressure. The spatial NAO patterns in the midHolocene and lig127k experiments are indistinguishable
55   from the piControl. The meridional temperature gradients weaken in the midHolocene and 127k experiments compared to the piControl, slightly reducing the magnitude of the positive NAO pattern, but do not reverse its spatial structure (Section 5). Both experiments still exhibit a positive NAO-like pattern. We will revise the text to eliminate the inconsistency.

Line 352: The figure reference is incorrect. Fig. 2a shows temperature changes, not sea-level pressure. Moreover, the text
60   claims "models do not capture a positive NAO-like pattern," but Figs. 2c and 2f clearly display such a pattern. Please reconcile the text with the figures.

"models do not capture a positive NAO-like pattern" comes from the finding of Mauri et al. (2014) as cited in manuscript. We will revise the text during revision.

65    Figure 4: although the caption notes that "the horizontal locations of dots in a does not have any meaning", it's easier for readers to understand if the black dots are in the shaded side (right side) of the distribution, as in Figure 5.

Thanks for the suggestion. We will update the figure.

Section 6 title ("remote effects"): I am wondering if "remote effects" is a good title for section 6. It suggests a focus on
70    NAO impacts on remote climates, but much of the section (e.g., Line 349) discusses NAO amplitude changes under different mean-state backgrounds. A more precise title would improve clarity.

Thanks for the suggestion. We will produce a new title for section 6.

The visibility of the paper could be improved with clearer writing, particularly in the way paragraphs are structured. I noticed
75    that some paragraphs cover multiple topics, and topic sentences are not always clearly stated. For example, in the paragraph starting at line 192, the first sentence is intended as a topic sentence, but it is actually a technical statement that belongs in the methods section (and could be deleted here). Instead, the sentence at line 195 would serve much better as the topic sentence.Another example is the paragraph starting at line 250. It begins with the NAO pattern, then moves on to explained variance, and finally to the magnitude of the NAO index. It would be much clearer if this paragraph were split into separate
80    ones, each focused on a single topic, with a strong topic sentence each. That way, the subsequent discussion of these three aspects (a new paragraph in your manuscript) could be integrated into the corresponding new paragraphs.

Thanks for the suggestion. We will improve the writing.

Minor points:
85    Line 85-88 I suggest to remove the very vague description from "State-of-the-art" to "in different scenarios". Line 175 NAO pattern is a dipole pattern, with negative anomaly at the northern center of action, and positive anomaly at the southern center of action. Therefore, the pattern is always "positive" and the phase is given by the sign of the NAO index. if the EOF gives a "negative" pattern, then both the "eof" and the "pc" should be multiplied by -1. "simulated pattern shows a positive phase" is misleading and should be revised. Line 235 to be more concise, "uncertainty in producing the location of the jet stream, arising
90    from both model bias and internal variability". Line 294 I suggest to remove the vague sentence "The NAO is a variation in the atmospheric pressure differences between the Icelandic Low and the Azores High."

Many thanks for listing the points. We will revise the text as suggested during the revision.